# Influence of the Method of Fe Deposition on the Surface of Hydrolytic Lignin on the Activity in the Process of Its Conversion in the Presence of CO_2_

**DOI:** 10.3390/ijms24021279

**Published:** 2023-01-09

**Authors:** Artem A. Medvedev, Alexander L. Kustov, Daria A. Beldova, Konstantin B. Kalmykov, Mikhail Yu. Mashkin, Anastasia A. Shesterkina, Sergey F. Dunaev, Leonid M. Kustov

**Affiliations:** 1Chemistry Department, Moscow State University, 119992 Moscow, Russia; 2Institute of Ecotechnologies, National University of Science and Technology MISiS, 119071 Moscow, Russia; 3N. D. Zelinsky Institute of Organic Chemistry RAS, 119991 Moscow, Russia

**Keywords:** lignin, carbon materials, carbon dioxide, carbon monoxide, CO_2_ conversion, gasification

## Abstract

Hydrolytic lignin is one of the non-demanded carbon materials. Its CO_2_-assisted conversion is an important way to utilize it. The use of the catalysts prepared by metal deposition on the surface of hydrolytic lignin makes it possible to apply milder conditions of the conversion process with CO_2_ and to improve the economic indicators. The development of methods of deposition of the active phase is a problem of high importance for any heterogeneous catalytic processes. This work aimed at investigating the influence of the conditions of iron deposition on the surface of hydrolytic lignin on the process of CO_2_-assisted conversion of lignin. Different Fe precursors (Fe(NO_3_)_3_, FeSO_4_, Fe_2_(SO_4_)_3_), solvents (water, isopropanol, acetone, and ethanol), and concentrations of the solution were used; the properties of Fe/lignin composites were estimated by SEM, EDX, TEM, XRD methods and catalytic tests. All the prepared samples demonstrate a higher conversion compared to starting lignin itself in the carbon dioxide-assisted conversion process. The carbon dioxide conversion was up to 66% at 800 °C for the sample prepared from Fe(NO_3_)_3_ using a twofold water volume compared to incipient wetness water volume as a solvent (vs. 39% for pure lignin).

## 1. Introduction

The methods of the industrial applications of waste carbon materials are a problem of high importance in terms of ecology and energetic efficiency. Moreover, they can be a source of non-fossil carbon for the synthesis of a large number of carbon-based materials such as polymers, solvents, etc. [1,2]. Compared to the use of fossil carbon sources, the use of waste carbon materials as sources of energy carriers could reduce the carbon footprint.

Hydrolytic lignin is one of the large-scale produced non-demanded carbon materials, which due to its low reactivity is not widely used as a source of carbon feedstock for further processing [3,4,5,6]. Currently, the interest of researchers is focused on the development of various methods of lignin depolymerization: pyrolysis [7,8,9,10,11,12,13], gasification [14,15,16,17,18,19,20], depolymerization assisted by microwave radiation [21,22,23,24,25,26,27,28,29], catalytic hydrogenation [30,31], electrochemical methods [32], etc.

Gasification of carbon materials such as hydrolytic lignin at atmospheric pressure is a promising method of utilization, due to the relative simplicity of the method and thus low capital and operating costs. Carbon dioxide-assisted conversion of waste carbon materials is especially interesting due to the utilization of carbon dioxide, which is a greenhouse gas [33,34].

One of the products of the carbon dioxide-assisted conversion of carbon materials, is carbon monoxide (Equation (1)). By not being a product of terminal oxidation, CO is an interesting gas that can be utilized in the chemical syntheses of important products for the chemical industry, for example, hydrocarbons via the Fischer–Tropsch process and acetic acid via methanol carbonylation. It can be also used directly for generating electricity at gas turbine stations. That is why CO production is important to produce value-added products, thus its production is large [35]. The proposed mechanism of the gasification of lignin in supercritical water includes the cleavage of the bonds C–O–C followed by the bonds C–C and the opening of benzene rings, the cracking of intermediate radicals C_x_H_y_, which can further undergo the oxidation [36]. On the other hand, according to the DFT calculation data, it was proposed to consider the following mechanism of the gasification of the lignin dimers with the β-5 linage as a model molecule fragment: it was shown that initially, the depolymerization process occurs; then the monomers are transformed into phenols, formaldehyde and CO [37]. By the way, formaldehyde is unstable under the reaction conditions, and its further decomposition into CO and H_2_ can be revealed in real experiments [38].

The interaction of a carbon-containing material with carbon dioxide is an endothermic process (Equation (1)), as a result, a significant level of conversion can be achieved only at a temperature of about 1000 °C [39].
CO_2_ + C ⇄ 2CO(1)

One of the main goals is to achieve economic efficiency in the process of carbon dioxide conversion of hydrolytic lignin by using milder conditions for this process. Deposited catalysts prepared from the carbon-containing material are needed to lower the temperature of the process and thus reduce the operating costs of carbon dioxide conversion into carbon monoxide [21,23,40,41,42]. Compounds of alkali, alkaline-earth and transition metals possess catalytic activity in such processes. The deposited catalysts based on iron triad metals are of particular interest due to their low cost as well as their low toxicity.

Despite the fact that the use of deposited catalysts on the surface of a carbon-containing material makes it possible to perform the process under milder conditions, it is necessary to develop the methods of catalyst preparation in terms of metal deposition for each material separately because of the specific surface properties such as porosity, wettability, etc.

In this paper, various methods of iron deposition on the surface of hydrolytic lignin were investigated; the resulting materials were studied by the following physical and chemical methods: SEM-EDX, XRD and TEM, and the catalytic studies in the process of CO_2_ conversion. The method of impregnation with different iron salts (Fe(NO_3_)_3_, FeSO_4_, Fe_2_(SO_4_)_3_) using different solvents (water by incipient wetness and by the excess impregnation, isopropanol, acetone, ethanol) was examined. The main novelty of this work is related to revealing the correlation between the method of iron deposition onto hydrolytic lignin, the resulting structural properties and the catalytic activity of the materials.

## 2. Results and Discussion

The series of samples of hydrolytic lignin loaded with iron species was prepared by the method of an incipient wet impregnation and the impregnation with the excess of the solution. The used precursors were iron(II) nitrate or iron(III) nitrate, or iron(II) sulfate or iron(III) sulfate. Different solvents were applied: water, isopropanol, acetone, and ethanol. 

### 2.1. SEM and EDX Characterization

Microphotographs of all the samples and the detailed composition determined by the EDX method can be found in Appendix A. The generalized results of the EDX studies are presented in Table 1.

The standard deviations of the surface elemental composition values increased as follows: LN-acetone < LN-2H_2_O < LN-iPrOH < LN-1H_2_O < LN-ethanol << LS-II << LS-III. These might indicate that iron species are distributed on the surface of the materials not so uniformly in the case of using each solvent. It can be noticed that the samples prepared by iron nitrate solution impregnation tend to demonstrate a more uniform distribution of iron on the surface than the samples prepared from iron sulfate solutions. The samples made with sulfates show the least uniformity of iron distribution, as can be seen from the much higher average iron content on the surface and the high value of the standard deviation. Moreover, SEM images for these samples demonstrate the relatively large areas of iron deposition in the case of Fe(II) sulfate (Appendix A).

That is why the iron distribution on the surface is directly determined by the synthesis conditions: the solvent and the iron precursor. A more uniform distribution can be obtained by choosing iron nitrate as a precursor and iPrOH, acetone, ethanol, or H_2_O (the twofold amount compared with the capacity of the material) as a solvent. The possible reason for such behavior can result from the fact that lignin is a hydrophobic macromolecule because the large fraction of the monomer unit consists of a benzene ring (possibly substituted) and hydrocarbon side chain. In the series of the applied solvents, the polarities of the solvents differ, and the affinity of the solvent to the lignin surface and the ability to dissolve (to solvate) the iron salts presumably differ too. It results in a different rate of iron oxide species growth and so there are two main options: the small particles are formed but their amount is large, or the relatively large particles are presented on the surface but their number is much less. All the options in between are also possible.

The examples of the EDX spectra collected from the points with different iron content are presented in Figure 1.

### 2.2. Transmission Electron Microscopy

The prepared samples were additionally investigated by the TEM technique (Figure 2). As can be noticed, the synthetic procedure affects the morphology of the particles of the metal oxide phase. Despite the samples LN-acetone, LN-2H_2_O, and LN-iPrOH demonstrating the least SD, these samples contain relatively large particles of iron oxides. Only the sample LN-1H_2_O shows relatively small iron oxide particles. Since the particles of iron oxide are relatively large, it can be proposed that the number of such iron oxide particles is not so large. However, assuming the whole amount of Fe atoms in the sample is constant, the species of smaller particles are present on the surface of the samples in a large amount, but they cannot be seen by SEM or TEM techniques because of the resolution.

The sample after the thermal treatment demonstrates a large portion of small, crystallized areas in the TEM image; the crystallization under the investigated conditions of the catalytic reaction can be proposed.

### 2.3. XRD

The observed reflexes apparently are the reflexes of a quartz phase according to the ICDD card [46-1045]. The source of such a phase can be proposed to be in the nature of lignin or in the contamination of the samples during industrial production. Areas with a high silicon content are found by EDX on the surface of the samples LN-iPrOH and LN-acetone (Appendix A). The difference in quartz reflex intensities can be explained presumably by the difference in the phase amount in the samples. Except for this phase, the samples are amorphous but one sample LS-II apparently contains the xitieshanite phase (Figure 3), and no other iron-containing crystalline phase can be determined because of the overlapping in XRD patterns of the quartz phase and possible iron-containing phases; the formation of small, probably even monolayered iron oxide particles can be proposed. The formation of the xitieshanite phase can be attributed to the oxidation of iron(II) sulfate while slow drying under the air atmosphere. This is consistent with the elemental maps of the samples (Appendix A) for which no agglomeration of Fe on the surface can be seen.

### 2.4. The Catalytic Tests

The dependences of the carbon dioxide conversion at a temperature of 800 °C for Fe-containing catalysts obtained under various synthetic conditions are shown in Figure 4. The most catalytically active material is hydrolytic lignin impregnated with iron(III) nitrate using water as a solvent; the impregnation was carried out with a twofold excess compared to the incipient wetness capacity. This sample demonstrated the CO_2_ conversion of 66% at 800 °C. This material makes it possible to achieve a catalytic activity about two times higher than what was observed for the pure lignin sample (which demonstrated a 39% CO_2_ conversion also at 800 °C).

All impregnation methods and all precursors showed an increase in the carbon dioxide conversion relative to the initial lignin. The materials obtained by impregnation with a solution of iron nitrate from non-aqueous solvents and water (by incipient wetness) showed close conversion values (about 53–59%). The samples prepared using iron sulfates showed the lowest catalytic activity in the studied process (42–49%), this might possibly be a result of the partial modification of the surface with sulfate groups. To summarize the results of the catalytic tests, the CO_2_ conversion decreases as follows: LN-2H_2_O > LN-acetone ≈ LN-1H_2_O = LN-iPrOH > LN-ethanol > LS-II > LS-III > pure lignin. It is noticeable that the four most active samples are the samples that have the highest uniformity of iron distribution on the surface of the samples. It can be proposed that the conditions of the synthesis directly determine the iron distribution, which further governs the catalytic activity.

Thus, the use of a solution with a lower concentration of iron(III) nitrate makes it possible to obtain the best samples in terms of homogeneity, and to achieve the highest conversion in the process of carbon dioxide-assisted conversion of hydrolytic lignin. It might result from the difference in the affinity of the applied solvent to the lignin (mostly hydrophobic) surface and its ability to form solvates with iron salts. In the case of iron(II) sulfate, it can be seen that the resulting phase observed by XRD is the phase iron(III), because of iron(II) oxidation under air atmosphere. The size of particles observed by TEM seems not to be influenced by catalytic activity: the most active sample contains relatively large particles. Nevertheless, it can be also proposed that not all the particles have a large size, but also the very small particles unobservable by TEM are also presented on the surface. This observation is indirectly consistent with XRD data: we cannot see the high reflexes of iron-containing phases which could be found in the large particles of crystal phases that were presented en masse. If water is used as a solvent, the lower concentration is preferable: this observation is kind of counterintuitive because the higher concentration should provoke the formation of seed crystals in large amounts, but it was shown by TEM that the small particles were formed in the case of higher iron(III) nitrate concentration. It probably can be explained by the specificity of the surface of lignin in certain samples in terms of the internal inhomogeneity of the materials prepared from the biological raw materials. In the case of using the sulfates of iron(II) and (III) the negative effect can be attributed to the impact of a double negative charged anion on the iron deposition or to the strength of interaction of the cation with anion; the more charged the ions, the more the Coulomb interaction. The second explanation is consistent both with the EDX and the catalytic characterization results: the Fe^3+^-containing samples have both less activity and higher standard deviation in the surface composition, which can result from iron(III) sulfate tending to form a compact particle compared to iron(II) sulfate.

## 3. Materials and Methods

### 3.1. Materials

The following reagents were used in this work: Fe(NO_3_)_3_·9H_2_O (99%) from Acros, FeSO_4_·7H_2_O (99%) from Merck, Fe_2_(SO_4_)_3_·9H_2_O (99%) from Acros, propanol-2 (>99%), ethanol (>99%), acetone (>99%), urea (>95%), hydrolytic lignin. All the reagents were used as purchased without further purification. The hydrolytic lignin originated from the residues from the furfural pilot industrial plant from the wood of oak and elm according to the technical conditions BY 490822905.001-2015. In brief, the hydrolysis of the cellulose in the original raw material was conducted using 2% H_2_SO_4_ at the temperature 110–140 °C. The ash content was declared to be 10.1 wt. %. The CHNOS elemental composition is presented in Appendix A.

### 3.2. Methods

The elemental composition of the samples of starting hydrolytic lignin was investigated with a Perkin Elmer 2400 Series II CHNOS analyzer. The sample weight was 50 mg. The results of the elemental analysis can be found in Appendix A.

All the materials with deposited iron were examined by SEM with EDX using a Leo Supra 50VP scanning electron microscope under a low vacuum in a nitrogen atmosphere. EDX data were collected using an energy-dispersive spectrometer INCA Energy (Oxford Instruments, X-Max-80, Abingdon, UK).

Transmission electron microscopy studies of the materials were performed using a transmission electron microscope JEM-2100 JEOL (Tokyo, Japan).

A powder XRD diffraction pattern was obtained on a device Rigaku IV Ultra using CuKα radiation. The samples were examined in the region 2θ = 10–60° at a rate of 1 degree per minute. Before the examination, the samples were heated at 300 °C for an hour under the atmosphere of CO_2_.

The evaluation of the activities of the resulting materials in a gasification process was performed using a quartz flow-type reactor with an internal diameter of 8 mm under a CO_2_ pressure of 1 atm, temperature ramp was 10 °C per minute, the temperature range was 100–850 °C, the flow rate of CO_2_ was 30 mL per minute, WHSV for 5 wt. % of metal was 70.7 h^−1^. A Bronkhorst EL-FLOW SELECT F-111B gas flow controller was used to determine the gas flow rate. The catalyst loading was 1 g, and the particle size was 0.25–0.5 mm. The reaction gas products were analyzed using a Chromatek Crystal 5000 gas chromatograph with thermal conductivity detectors, M ss316 3 m × 2 mm columns, Hayesep Q 80/100 mesh, and CaA molecular sieves.

The following reactions (Equations (1)–(3)) take place in the reaction zone.
CO + H_2_O → CO_2_ + H_2_(2)
2H_2_ + C → CH_4_(3)

The conversion (X_CO_2__) of carbon dioxide during the tests was calculated by the formula (Equation (4)).
(4)$$X_{{\mathrm{CO}}_{2}}=\frac{0.5n\left(\mathrm{CO}\right)+n\left({\mathrm{CH}}_{4}\right)}{n\left({\mathrm{CO}}_{2}\right)+0.5n\left(\mathrm{CO}\right)-2n\left({\mathrm{CH}}_{4}\right)}$$

### 3.3. Synthetic Procedure

The deposition of iron compounds on hydrolytic lignin was carried out by the incipient wetness impregnation method. The mass fraction of deposited metal in all samples was adjusted to be 5 wt. %. Solvents (distilled water, ethanol, propanol-2, acetone), iron salt anions (nitrate and sulfate), the iron oxidation states for sulfates (iron(II) sulfate, iron(III) sulfate), the concentrations of iron(III) nitrate in the impregnating solution were also varied (using distilled water exactly corresponding to the incipient wetness capacity and in a twofold excess). The moisture capacities of lignin for each solvent are summarized in Appendix A.

All the samples were prepared as follows: the selected iron salt was completely dissolved in the appropriate amount of the selected solvent. The amount of hydrolytic lignin (2 g) was impregnated with the appropriate salt solution. The samples were dried at room temperature for about a week to obtain the dry samples.

The resulting materials were denoted as LN-1H_2_O and LN-2H_2_O for the samples impregnated with onefold and twofold volumes of the solution of iron(III) nitrate; LN-iPrOH, LN-acetone, LN-ethanol for the samples impregnated with the solution of iron(III) nitrate in each of the organic solvents; LS-II and LS-III for the samples prepared from sulfates of iron(II) and iron(III) in a water solution, respectively.

## 4. Conclusions

The method of impregnation with iron(III) nitrate from an aqueous solution with a twofold excess of water relative to the incipient wetness showed the highest homogeneity of iron distribution on the surface of lignin, and the closest value of the iron content on the surface to the target value. An additional advantage of this application method is the use of a “green” solvent—water.

Studies of the catalytic activity of the obtained materials in the process of carbon dioxide-assisted conversion of lignin showed that the most active material is hydrolytic lignin impregnated with iron(III) nitrate using water as a solvent in a twofold excess relative to the incipient wetness (the CO_2_ conversion was 66% at 800 °C). Thus, the uniformity of the distribution of iron nanoparticles determines the conversion of carbon dioxide in the reaction of CO_2_-assisted conversion of hydrolytic lignin.

## Figures and Tables

**Figure 1 ijms-24-01279-f001:**
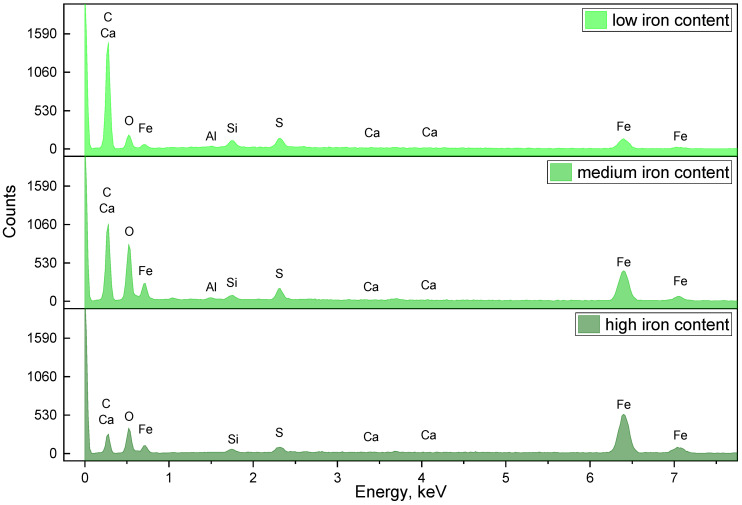
The EDX spectra for the points with low, medium and high iron content.

**Figure 2 ijms-24-01279-f002:**
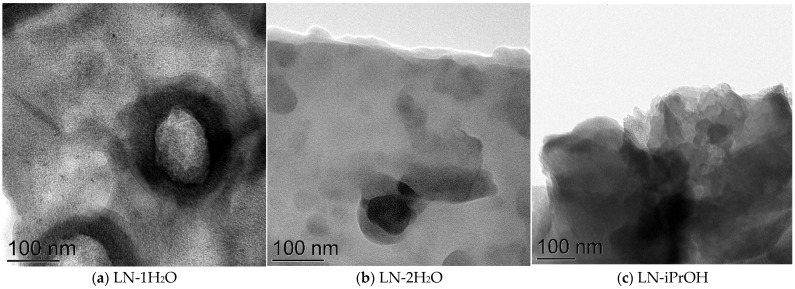
TEM images for the prepared materials under the chosen synthesis conditions and the sample LN-2H_2_O after 500 °C treatment.

**Figure 3 ijms-24-01279-f003:**
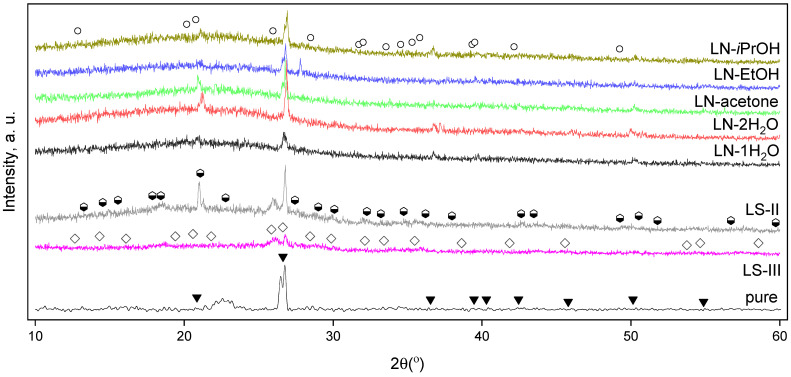
The diffractograms of the samples of hydrolytic lignin impregnated with different iron salts solutions and of the starting lignin. The ICDD patterns of quartz [46-1045] (B), fibroferrite (iron(III) sulfate hydroxide pentahydrate) [38-0481] (M), xitieshanite (iron(III) sulfate hydroxide heptahydrate) [35-0719] (ω), iron(III) nitrate nonahydrate [01-0124] (−) are shown to indicate the locations of the possible phases reflexes.

**Figure 4 ijms-24-01279-f004:**
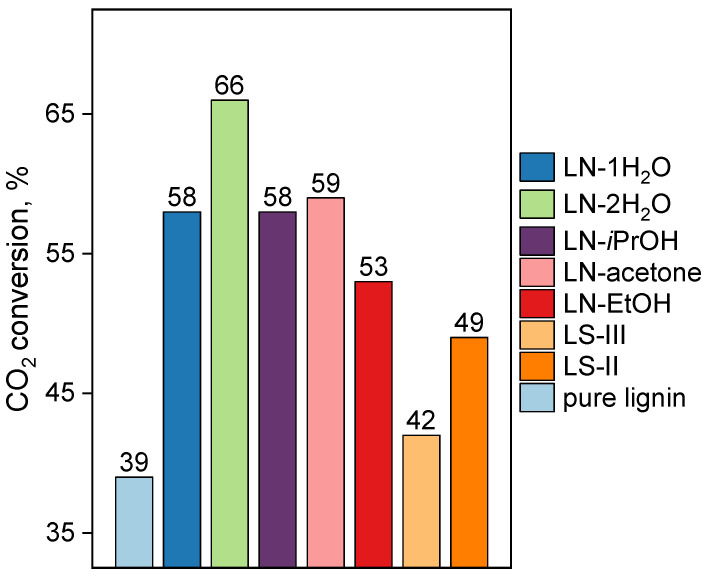
The results of the catalytic tests for the prepared Fe-containing materials and the starting lignin in the reaction of the conversion of lignin in the presence of CO_2_ at 800 °C.

**Table 1 ijms-24-01279-t001:** Average values and standard deviations of the iron content on the surface of the impregnated materials.

Sample	Average Fe Content, wt. %	SD
LN-1H_2_O	7.8	2.1
LN-2H_2_O	5.8	1.5
LN-iPrOH	5.5	2.0
LN-acetone	6.1	1.3
LN-ethanol	6.6	2.7
LS-III	9.9	13.7
LS-II	9.7	9.9

## Data Availability

Data are available from the authors upon request.

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
