# Peer review of "Influence of the Method of Fe Deposition on the Surface of Hydrolytic Lignin on the Activity in the Process of Its Conversion in the Presence of CO2"

_ijms, 2023, doi:10.3390/ijms24021279_

Round 1
Reviewer 1 Report
The authors have reported the use of the catalysts prepared by metal deposition on the surface of hydrolytic lignin via different precursors and solvents. The material is characterized by SEM-EDS, TEM, and XRD and its catalytic activity is monitored via CO2-assisted conversion. The manuscript requires a lot of work as it lacks sufficient discussion and reasoning to convey the results and discussion. I do not recommend the paper for publication:
1) Line 45-47: It is important to emphasize and expand the discussion of generating CO as a product.
2) Line 77: Please summarize the preparation of the materials at the beginning before discussing the results despite being mentioned in the material and method section.
3) Line 83-85: “The standard deviations of the surface elemental composition values were increasing as following: LN-acetone < LN-2H2O < LN-iPrOH < LN-1H2O < LN-ethanol << LS-II << LS-84 III.”
What are these conclusions based on? No figure numbers are marked.
4) Line 91: Again, please mention the figure number this observation is based on. Also, I would suggest including the EDS analysis next to their corresponding SEM images for easier understanding in the main manuscript.
5) Line 93-96: Need to provide possible reasoning on the effect of the solvent and precursor over the observed material/catalyst.
6) Line 105-108: Please rephrase the sentence as it is very confusing.
7) Line 116: “The observed reflexes apparently are the reflexes of a quartz phase.”
No citations or references for this observation. Also, need to include background XRD spectra of lignin.
8) Line 121: “and no iron- 121 containing crystalline phase can be found,”.
What peaks correspond to Iron in XRD? Std XRD spectra for Iron need to be included.
9) Equation 4: There should be a decimal instead of a comma in 0.5n(CO)
Author Response
We sincerely thank our reviewers for their valuable comments and their suggestions on improving our manuscript. All references were checked to be relevant to the contents of the manuscript. The revisions made in the manuscript were marked with blue colour to highlight the changes. The English language was re-checked and some typos and grammar mistakes were addressed. Below we provide the comments of the reviewers with our responses to each of them.
Reviewer 1
- Line 45-47: It is important to emphasize and expand the discussion of generating CO as a product.
The discussion on the mechanism of the process including CO generation was enlarged and now it looks like following.
Not being a product of the terminal oxidation, CO is an interesting gas which can be utilized in many chemical syntheses of important products for the chemical industry, for example, hydrocarbons by Fischer-Tropsch process and acetic acid by methanol carbonylation. It can be also used directly for generating electricity at gas turbine stations. That is why CO production is an important way to production of the row of the value-added products and its production is large [35]. The proposed mechanism of the gasification of lignin in supercritical water includes firstly the cleavage of the bonds C–O–C, further the bonds C–C and the opening of benzene rings, cracking: the intermediate radicals CHx, which can further undergo the oxidation [36]. On the other hand, according to the DFT calculational data it was proposed to consider the following mechanism of the gasification of the lignin dimers with b-5 linage as a model molecule fragment: it was shown that initially the depolymerization process occurs and further the monomers are transformed into phenols, formaldehyde and CO [37]. By the way, formaldehyde is unstable under the reaction conditions, and its further decomposition into CO and H2 can be seen in the real experiments [38].
2) Line 77: Please summarize the preparation of the materials at the beginning before discussing the results despite being mentioned in the material and method section.
The preparation of the materials was summarised in the beginning of the section Results and Discussion as follows.
The series of the samples of the hydrolytic lignin loaded with iron species was prepared by the method of an incipient wet impregnation and the impregnation with the excess of the solution. The used precursors were iron (II) nitrate or iron (III) nitrate, or iron (II) sulfate or iron (III) sulfate. The different solvents were applied: water, isopropanol, acetone, ethanol.
3) Line 83-85: “The standard deviations of the surface elemental composition values were increasing as following: LN-acetone < LN-2H2O < LN-iPrOH < LN-1H2O < LN-ethanol << LS-II << LS-84 III.”
What are these conclusions based on? No figure numbers are marked.
The line 78 now contains the reference to the Supporting Information, Figures S1–S14, Tables S3–S9. The detailed results of the EDX characterisation were placed in Supporting Information because they would enlarge the manuscript significantly, but their value is only to be the raw data. So, we believe that the summarising table (Table 1 in the manuscript) is enough to show the EDX results. All the interested readers can find the figures in the provided Supporting Information file.
4) Line 91: Again, please mention the figure number this observation is based on. Also, I would suggest including the EDS analysis next to their corresponding SEM images for easier understanding in the main manuscript.
The observation is based on Supporting Information, Figure S13. This figure follows. The white areas correspond to the iron-rich phase according to the EDX data.
|
|
Figure S13. Microphotos of the sample LS-II. EDX data were collected from the marked areas.
5) Line 93-96: Need to provide possible reasoning on the effect of the solvent and precursor over the observed material/catalyst.
The possible reason of such behavior can result from the fact that lignin is a hydrophobic macromolecule because of the large fraction of the monomer unit consists of a benzene ring (possibly substituted) and hydrocarbon side chain. In the series of the applied solvents the polarities of the solvents differ, the affinity of the solvent to the lignin surface and the ability to dissolve (to solvate) the iron salts presumably differs too. It results in a different rate of iron oxide species growth and so there are two main options: the small particles are formed but their amount is large, or the relatively large particles are presented on the surface but its number is much less. All the options in between are also possible.
6) Line 105-108: Please rephrase the sentence as it is very confusing.
The sentence was rephrase as presented below.
Since the particles of iron oxide are relatively large, it can be proposed that the number of such iron oxide particles is not so large. But assuming the whole amount of Fe atoms in the sample is a constant, the species of smaller particles are present on the surface of the samples in a large amount, but they cannot be seen by SEM or TEM techniques because of the resolution abilities.
7) Line 116: “The observed reflexes apparently are the reflexes of a quartz phase.”
No citations or references for this observation. Also, need to include background XRD spectra of lignin.
The ICDD XRD patterns of possible iron-containing phases were shown in Figure 2 in the manuscript (the picture below). The XRD pattern of pure lignin was also added.
Figure 2. The diffractograms of the samples of hydrolytic lignin impregnated with different iron salts solutions and of the starting lignin. The ICDD patterns of quartz [46-1045] (B), fibroferrite (iron(III) sulfate hydroxide pentahydrate) [38-0481] (M), xitieshanite (iron(III) sulfate hydroxide heptahydrate) [35-0719] (v), iron(III) nitrate nonahydrate [01-0124] (-) are shown to indicate the locations of the possible phases reflexes.
8) Line 121: “and no iron- 121 containing crystalline phase can be found,”.
What peaks correspond to Iron in XRD? Std XRD spectra for Iron need to be included.
The XRD patterns of the possible phase mostly like to the observed reflexes are now presented. The section of XRD now looks as below.
The observed reflexes apparently are the reflexes of a quartz phase according to the ICDD card [46-1045]. The source of such a phase can be proposed to be in the nature of lignin or in the contamination of the samples during industrial production. Areas with a high silicon content are found by EDX on the surface on the samples LN-iPrOH and LN-acetone (SI, Figures S6, S8). The difference in quartz reflexes intensities can be explained presumably by the difference in the phase amount in the samples. Except this phase, the samples are amorphous but one sample LS-II apparently containing the xitieshanite phase, and no other iron-containing crystalline phase can be determined because of the overlapping in XRD patterns of the quartz phase and possible iron-containing phases, and the formation of small, probably even monolayered, iron oxide particles can be proposed. The formation of xitieshanite phase can be attributed to the oxidation of iron(II) sulfate while slow drying under the air atmosphere. This is consistent with elemental maps of the samples (Supporting Information, Figures S2, S4, S8, S10, S12) for which no agglomeration of Fe on the surface can be seen.
9) Equation 4: There should be a decimal instead of a comma in 0.5n(CO)
The appropriate correction was performed in the manuscript.

Reviewer 2 Report
This manuscript deals with an interesting gasification procedure under iron catalysis. However, it shows some flaws that require Authors' intervention. In particular, the Materials & Methods section is too concise, and the reader is forced to read other sources to find some essential information. "Hydrolytic lignin" could means several different preparations, and should be detailed adequately. The drying procedure for the samples prior to gasification experiments is not described at all. Moreover, the "Results and Discussion" section in fact does not contain any discussion, and merely presents (interesting) experimental data that would deserve a minimum real discussion. E.g., do the Authors have a tentative hypothesis/explanation for the gasification conversion pattern they have observed depending on each individual lignin/iron preparation they have tested? In the present form, the article looks rather like a short communication or a letter to the Editor than a complete research article. In my opinion, a (deep) Authors' intervention on the manuscript would largely improve it...
Last but not least, I was unable to download the Supplementary Information ("Error 404 - file not found")
Author Response
We sincerely thank our reviewers for their valuable comments and their suggestions on improving our manuscript. All references were checked to be relevant to the contents of the manuscript. The revisions made in the manuscript were marked with blue colour to highlight the changes. The English language was re-checked and some typos and grammar mistakes were addressed. Below we provide the comments of the reviewers with our responses to each of them.
Reviewer 2
This manuscript deals with an interesting gasification procedure under iron catalysis. However, it shows some flaws that require Authors' intervention.
In particular, the Materials & Methods section is too concise, and the reader is forced to read other sources to find some essential information.
The section Materials and Methods were enlarged with the information on the origin of the used lignin, the pre-treatment for the XRD examination, the details of the catalytic evaluation conditions and the detection of the products of the gasification process. The drying conditions was added to the Synthetic procedure subsection.
"Hydrolytic lignin" could mean several different preparations, and should be detailed adequately.
The hydrolytic lignin originated from the residues from the furfural pilot industrial plant from the wood of oak and elm according to the technical conditions BY 490822905.001-2015. In brief, the hydrolysis of the cellulose in the original raw material were conducted using 2% H2SO4 at the temperature 110–140 °C. The ash content was declared to be 10.1 wt. %. The CHNOS elemental composition are presented in Supporting Information, Table S1.
The drying procedure for the samples prior to gasification experiments is not described at all.
The samples were dried at room temperature for about a week to obtain the dry powder of a composite material.
Moreover, the "Results and Discussion" section in fact does not contain any discussion, and merely presents (interesting) experimental data that would deserve a minimum real discussion. E.g., do the Authors have a tentative hypothesis/explanation for the gasification conversion pattern they have observed depending on each individual lignin/iron preparation they have tested? In the present form, the article looks rather like a short communication or a letter to the Editor than a complete research article. In my opinion, a (deep) Authors' intervention on the manuscript would largely improve it...
The discussion of the results were enlarged in terms to propose a hypothesis why the behavior of the samples differs from each other.
Thus, the use of a solution with a lower concentration of iron(III) nitrate makes it possible to obtain the best samples in terms of homogeneity, and to achieve the highest conversion in the process of carbon dioxide assisted conversion of hydrolytic lignin. It possibly might result from the difference in the affinity of the applied solvent to lignin (mostly hydrophobic) surface and its ability to form solvates with iron salts. In case of iron(II) sulfate it can be seen that the resulting phase observed by XRD is the phase if iron(III) because of iron(II) oxidation under air atmosphere. The size of particles observed by TEM seems not to be influencing on the catalytic activity: the most active sample contains relatively large particles. Nevertheless, it can be also proposed that not all the particles have a large size, but also the very small, unobservable by TEM, particles are also presented on the surface. This observation is indirectly consistent with XRD data: we cannot see the high reflexes of iron-containing phases which could be found if the large particles of crystal phases were presented en masse. If water used as a solvent, the lower concentration is preferable: this observation is kind of counterintuitive because the higher concentration should provoke the formation of seed crystals in large amounts, but it was shown by TEM that the small particles were formed in the case of higher iron(III) nitrate concentration. Probably it can be explained by the specificity of the surface of lignin in a certain samples in terms of the internal inhomogeneity of the materials prepared from the biological raw materials. In the case of using the sulfates of iron(II) and (III) the negative effect can be attributed to the impact of double negative charged anion on the iron deposition or to the strength of interaction of cation with anion and the more charged the ions, the more the Coulomb interaction. The second explanation is consistent both with the EDX and the catalytic characterization results: the Fe3+-containing samples have both less activity and higher standard deviation in the surface composition, which can result from that iron(III) sulfate more tends to form a compact particle being compared to iron(II) sulfate.
Last but not least, I was unable to download the Supplementary Information ("Error 404 - file not found")
The file Supporting Information was provided. In case of any troubles, we can additionally send it, for example, by e-mail.

Round 2
Reviewer 1 Report
The authors have incorporated the provided suggestion and improvised the manuscript. I recommend it for publication now.